# Improving Benign and Malignant Classifications in Mammography with ROI-Stratified Deep Learning

**DOI:** 10.3390/bioengineering12080885

**Published:** 2025-08-20

**Authors:** Kenji Yoshitsugu, Kazumasa Kishimoto, Tadamasa Takemura

**Affiliations:** 1Graduate School of Information Science, University of Hyogo, 7-1-28 Minatojima Minamimachi, Chuo-ku, Kobe-shi 650-0047, Hyogo, Japan; takemura@ai.u-hyogo.ac.jp; 2Division of Medical Information Technology and Administration Planning, Kyoto University Hospital, 54 Kawaramachi, Shogoin, Sakyo-ku 606-8507, Kyoto, Japan; kishimoto@kuhp.kyoto-u.ac.jp

**Keywords:** deep learning, mammography, region of interest

## Abstract

Deep learning has achieved widespread adoption for medical image diagnosis, with extensive research dedicated to mammographic image analysis for breast cancer screening. This study investigates the hypothesis that incorporating region-of-interest (ROI) mask information for individual mammographic images during deep learning can improve the accuracy of benign/malignant diagnoses. Swin Transformer and ConvNeXtV2 deep learning models were used to evaluate their performance on the public VinDr and CDD-CESM datasets. Our approach involved stratifying mammographic images based on the presence or absence of ROI masks, performing independent training and prediction for each subgroup, and subsequently merging the results. Baseline prediction metrics (sensitivity, specificity, F-score, and accuracy) without ROI-stratified separation were the following: VinDr/Swin Transformer (0.00, 1.00, 0.00, 0.85), VinDr/ConvNeXtV2 (0.00, 1.00, 0.00, 0.85), CDD-CESM/Swin Transformer (0.29, 0.68, 0.41, 0.48), and CDD-CESM/ConvNeXtV2 (0.65, 0.65, 0.65, 0.65). Subsequent analysis with ROI-stratified separation demonstrated marked improvements in these metrics: VinDr/Swin Transformer (0.93, 0.87, 0.90, 0.87), VinDr/ConvNeXtV2 (0.90, 0.86, 0.88, 0.87), CDD-CESM/Swin Transformer (0.65, 0.65, 0.65, 0.65), and CDD-CESM/ConvNeXtV2 (0.74, 0.61, 0.67, 0.68). These findings provide compelling evidence that validate our hypothesis and affirm the utility of considering ROI mask information for enhanced diagnostic accuracy in mammography.

## 1. Introduction

For breast cancer, which remains a prevalent malignancy among women worldwide, early detection and accurate diagnosis are crucially important for improving survival. Mammography is the widely adopted standard for breast cancer screening, but its interpretation demands extensive expertise. Challenges persist related to diagnostic discrepancies and missed diagnoses among radiologists.

Beyond mammography, breast cancer diagnosis incorporates various methods, including visual inspection, palpation, and ultrasound examination. When these examinations reveal abnormalities, clinicians often perform highly invasive procedures such as cytological and histological examinations for definitive diagnosis. If deep learning-based image analysis of minimally invasive mammographic images were to achieve high diagnostic accuracy, then the need for highly invasive procedures would be reduced. This approach would simultaneously alleviate burdens on radiologists and breast surgeons responsible for interpreting these images.

Recent rapid advancements in artificial intelligence (AI) technology, particularly deep learning, have significantly accelerated the development of automated analysis and diagnostic support systems for mammographic images. For various image recognition tasks, deep learning algorithms, especially convolutional neural networks (CNNs), now demonstrate performance comparable to or exceeding human capabilities. For medical image diagnosis, these technologies often achieve superior accuracy and efficiency compared to conventional methodologies.

Many studies have explored deep learning applications for mammographic image diagnosis. For instance, Zhang et al. [1] performed a two-stage classification (normal/abnormal and benign/malignant) using two-view mammograms (CC and MLO) on the public DDSM dataset using a multi-scale attention DenseNet. Lång et al. [2] evaluated the potential of AI to identify normal mammograms by classifying cancer likelihood scores with a deep learning model on a private dataset, comparing the obtained results to radiologists’ interpretations. Another study by Lång et al. [3] indicated that deep learning models trained on a private dataset can reduce cancer rates at intervals without supplementary screening. Zhu et al. [4] predicted future breast cancer development in negative subjects during an eight-year period using a deep learning model with a private dataset. Kerschke et al. [5] compared human versus deep learning AI accuracy for benign–malignant screening using a private dataset, highlighting the need for prospective studies. Nica et al. [6] reported a high-accuracy benign–malignant classification of cranio-caudal view mammography images using an AlexNet deep learning model and a private dataset. Rehman et al. [7] achieved high-accuracy architectural distortion detection using image processing and proprietary depth-wise 2D V-net 64 convolutional neural networks on the PINUM, CBIS-DDSM, and DDSM datasets. Yirgin et al. [8] used a public deep learning diagnostic system on a private dataset, concluding that combined assessment with both the deep learning model and radiologists yielded the best performance. Tzortzis et al. [9] demonstrated superior performance for efficiently detecting abnormalities on the public INBreast dataset using their tensor-based deep learning model, showing robustness with limited data and reduced computational requirements. Pawar et al. [10] and Hsu et al. [11] both reported high-accuracy Breast Imaging Reporting and Data System (BIRADS) category classifications, respectively, using proprietary multi-channel DenseNet architecture and a fully convolutional dense connection network on private datasets. Elhakim et al. [12] investigated the feasibility of replacing the first reader with AI when double-reading mammography using a commercial AI system with a private dataset, emphasizing the importance of an appropriate AI threshold. Jaamour et al. [13] improved the segmentation accuracy for mass and calcification images from the public CBIS-DDSM dataset by applying transfer learning. Kebede et al. [14] developed a model combining EfficientNet-based classifiers with a YOLOv5 object detection model and an anomaly detection model for mass screening on the public VinDr and Mini-DDSM datasets. Ellis et al. [15], using the UK national OPTIMAM dataset, developed a deep learning AI model for predicting future cancer risk in patients with negative mammograms. Elhakim et al. [16] further investigated the replacement of one or both readers with AI when double-reading mammography images, emphasizing clinical implications for accuracy and workload. Sait et al. [17] reported high segmentation accuracy and generalizability in multi-class breast cancer image classification using an EfficientNet B7 model within a LightGBM model on the CBIS-DDSM and CMMD datasets. Chakravarthy et al. [18] reported high classification accuracy for normal, benign, and malignant cases using an ensemble method with a modified Gompertz function on the BCDR, MIAS, INbreast, and CBIS-DDSM datasets. Liu et al. [19] achieved high classification accuracy on four binary tasks using a CNN and a private mammography image dataset, suggesting the potential to reduce unnecessary breast biopsies. Finally, Park et al. [20] reported improved diagnostic accuracy, especially in challenging ACR BIRADS categories 3 and 4 with breast density exceeding 50%, learning both benign–malignant classification and lesion boundaries using a ViT-B DINO-v2 model on the public CBIS-DDSM dataset. AlMansour et al. [21] reported high-accuracy BIRADS classification using MammoViT, a novel hybrid deep learning framework, on a private dataset.

Despite these advancements, several points of difficulty hinder the reproducibility of claims in deep learning applications for mammographic image diagnosis. Studies using private, non-public datasets or proprietary deep learning models with undisclosed details make verification difficult. Methods incorporating subject information alongside mammographic images as training data also face reproducibility issues caused by limited commonalities across different datasets. Similarly, studies combining mammographic images with other modality images require specific data combinations, thereby complicating claim reproduction.

Given these considerations, we prioritized reproducible research by addressing studies using publicly available datasets and open-source deep learning models. Furthermore, we emphasized the generalizability of claims across multiple public datasets and various deep learning models.

Scrutiny of the dataset indicated that the number of regions of interest (ROIs) tends to increase along with symptom severity: normal, benign, and malignant. This tendency suggests that the presence or absence of ROIs is a useful feature.

Therefore, this study tested the hypothesis that prediction accuracy improves when images are classified based on whether or not they have annotated mask information for regions of interest, with subsequent separate training and prediction for each of the four mammographic views (RCC, LCC, RMLO, LMLO), before merging the results. A standard mammographic examination typically includes four views: the left mediolateral oblique view (LMLO), right mediolateral oblique view (RMLO), left craniocaudal view (LCC), and right craniocaudal view (RCC) (Figure 1).

This approach is compared to cases for which image data are not separated based on the availability of mask information for regions of interest. Using two public datasets and two deep learning models, we validated this hypothesis, addressing the presence or absence of annotated mask information as a novel feature.

## 2. Materials and Methods

### 2.1. Materials

This study used two publicly available mammography datasets with region of interest (ROI) annotations: VinDr [22] and Categorized Digital Database for Low-Energy and Subtracted Contrast-Enhanced Spectral Mammography (CDD-CESM) [23]. Both datasets include ROI mask information, but not all mammographic images within them have corresponding mask images available.

During the training and prediction phases, our study exclusively considered the presence or absence of corresponding ROI images for individual mammographic images. The ROI images themselves were not used as input data.

The VinDr dataset provides BI-RADS information, but it lacks explicit benign–malignant classifications. Consequently, images categorized as BI-RADS 2 and 3 were classified as benign lesions, whereas those categorized as BI-RADS 4 and 5 were classified as malignant.

The CDD-CESM dataset includes predefined normal, benign, and malignant classifications. For this analysis, we used benign and malignant data exclusively.

Because the CDD-CESM dataset does not provide a predefined train–test split, we divided the data into training and testing sets with a 10:1 ratio.

Compositions of the respective datasets are presented in Table 1, Table 2, Table 3 and Table 4.

### 2.2. Methods

For this study, we adopted the following approach: initially, mammography images were classified based on the presence or absence of an ROI; they were then differentiated as either benign or malignant. Figure 2 presents an overview of the proposed method.

#### 2.2.1. Image Preprocessing

For preprocessing, window processing was applied during the conversion of DICOM images to the JPEG format. Windowing, in the context of DICOM imaging, is a process of mapping a specific range of pixel values from an image with a wide dynamic range, such as a mammogram, onto the display’s grayscale range (typically 0–255). This mapping is determined specifically according to two parameters stored within the DICOM file, Window Center (WC) and Window Width (WW), which together define the range of pixel values to be displayed. This technique enhances the visual perception of the image by allowing for the adjustment of contrast and brightness, thereby improving the visibility of relevant anatomical structures.

This preprocessing was followed by contrast adjustments using Contrast-Limited Adaptive Histogram Equalization (CLAHE).

#### 2.2.2. Image Classification Models

The Swin Transformer [24] and ConvNeXtV2 [25] were selected as image classification models because of their superior performance among the various models evaluated.

#### 2.2.3. Validation Procedure

The validation procedures were identical for both image classification models, involving the following steps:Mammographic images were segregated based on the presence or absence of ROI mask images.Images were divided into four standard views: right craniocaudal (RCC), left craniocaudal (LCC), right mediolateral oblique (RMLO), and left mediolateral oblique (LMLO).Training was performed on mammographic images without ROI mask images, with separate training undertaken for each view.Prediction was performed on mammographic images without ROI mask images, with separate predictions for each view.Training was then performed on mammographic images with ROI mask images, with separate training for each view.Prediction was performed on mammographic images with ROI mask images, with separate predictions for each view.Finally, the prediction results were merged.

#### 2.2.4. Comparative Validation Procedure

The comparative validation procedure differed from primary validation in that the presence or absence of ROI mask images was not considered during processing. The steps used for this procedure were the following.

Mammographic images were divided into the four standard views: RCC, LCC, RMLO, and LMLO.Training was performed on mammographic images without ROI mask images for each view.Prediction was performed on mammographic images without ROI mask images for each view.Training was performed on mammographic images with ROI mask images for each view.Prediction was performed on mammographic images with ROI mask images for each view.The prediction results were merged.

We used sensitivity, specificity, F-score, and accuracy as the evaluation metrics. These four evaluation metrics are defined by Equations (1)–(4), where TP, FP, TN, and FN respectively denote True Positives, False Positives, True Negatives, and False Negatives.Sensitivity = TP/(TP + FN)(1)Specificity = TN/(TN + FP)(2)F-score = 2(TP/(TP + FP) × Sensitivity))/(TP/(TP + FP) + Sensitivity)(3)Accuracy = (TP + TN)/(TP + FP + TN + FN)(4)

#### 2.2.5. Training Hyperparameters and Computational Environment

Training hyperparameters were determined through five-fold cross-validation. They remained consistent for both image classification models with a learning rate of 0.0001, 100 epochs, and an image size of 384 × 384 pixels. All other hyperparameters were maintained at their respective default values for each model.

Validation was conducted on a system running Windows 11 Pro, equipped with a 13th Gen Core (TM) i9-13900KF 3.00 GHz processor (Intel Corp., Santa Clara, CA, USA), 128 GB of memory, and an RTX 3090 GPU (NVIDIA Corp., Santa Clara, CA, USA).

## 3. Results

Our mammographic image classification results are presented for two scenarios: with and without the inclusion of ROI mask images. For the VinDr dataset, the classification results obtained using Swin Transformer and ConvNeXtV2 are shown in Table 5 and Table 6, respectively. Similarly, for the CDD-CESM dataset, Table 7 and Table 8 present the classification results obtained using Swin Transformer and ConvNeXtV2, respectively.

Although the Swin Transformer outperformed ConvNeXtV2 on both datasets, both models demonstrated improved classification performance compared to the baseline without considering ROIs.

## 4. Discussion

Our “ROI-Stratified” approach emphasizes data stratification based on the binary presence or absence of an ROI. This strategy shows promise, but it does not leverage richer, quantitative information about the ROIs, such as their size, shape, and internal texture. Considering that diagnosticians use these features as diagnostic cues, our current binary treatment might oversimplify the available information. A key challenge for future work, therefore, is to explore methods that incorporate these quantitative ROI features as additional inputs into the model, potentially facilitating a more nuanced decision-making process.

Some data lead to a diagnosis that is malignant but lacks a visible region of interest (ROI). This discrepancy is likely attributable to factors such as dense breast tissue, which can obscure ROIs by causing the entire image to appear uniformly opaque. In such cases, a malignant diagnosis is reached despite the absence of a clear ROI on the image, which will likely necessitate corroborating results from other diagnostic modalities such as biopsies. Instances where data was diagnosed as normal but exhibited an ROI were also observed. The presence of an ROI in a “normal” diagnosis seems unusual and suggests a potential misrepresentation or artifact in the diagnostic labeling process. Such anomalous data points, whether they involve a malignant diagnosis without a discernible ROI or a normal diagnosis with an ROI, introduce noise into deep learning models. This noise can strongly hinder the model’s ability to learn accurate patterns. The noise consequently diminishes its predictive performance. Preprocessing the dataset to identify and remove or re-evaluate these inconsistent data points before training might enhance the learning and prediction accuracy of deep learning algorithms for medical image analysis.

This study used data with pre-existing ROI mask images. However, mammographic images requiring a benign–malignant classification do not always have corresponding mask images available. Therefore, future research should specifically examine the generation of mask images for mammographic data lacking existing masks, employing techniques such as semantic segmentation or object detection, and subsequently validating these approaches.

The deep learning models used for this study, such as Swin Transformer and ConvNeXtV2, demonstrated superior accuracy in both training and prediction compared to other deep learning models. We hypothesize that this improved performance derives from differences in the respective layer architectures of these models. Detailed analysis of this phenomenon is a subject for future investigation.

Other architectures, including those already existing and some yet to be developed, might outperform the two models used for this study. Consequently, an exhaustive investigation into a broader range of high-performance models remains a key avenue for future research.

While this study specifically addressed benign–malignant classifications, mammographic data are typically categorized into normal versus abnormal findings, with abnormal cases subsequently classified as either benign or malignant. An important area for future investigation is assessing whether our methodology can effectively classify normal and abnormal cases. If successful, then this capability would enable diagnostic prediction for a broader range of arbitrary mammographic data.

## 5. Conclusions

Our findings show that pre-segmentation of mammographic data, based on the presence or absence of ROI mask images, the separation of training and prediction processes for each segment, and merging the results, can enhance classification accuracy for the benign–malignant classification of mammographic images using deep learning.

Furthermore, we demonstrated the generalizability of our proposed method by evaluating it using two public datasets and two deep learning models.

## Figures and Tables

**Figure 1 bioengineering-12-00885-f001:**
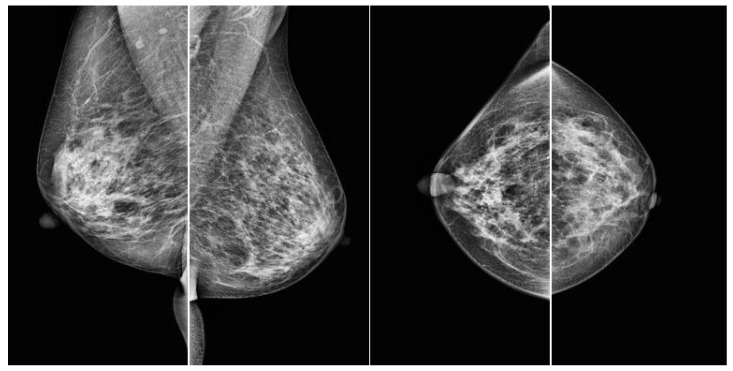
Samples of four views from a mammography from VinDr [22]. From left to right: LMLO, RMLO, LCC, and RCC.

**Figure 2 bioengineering-12-00885-f002:**
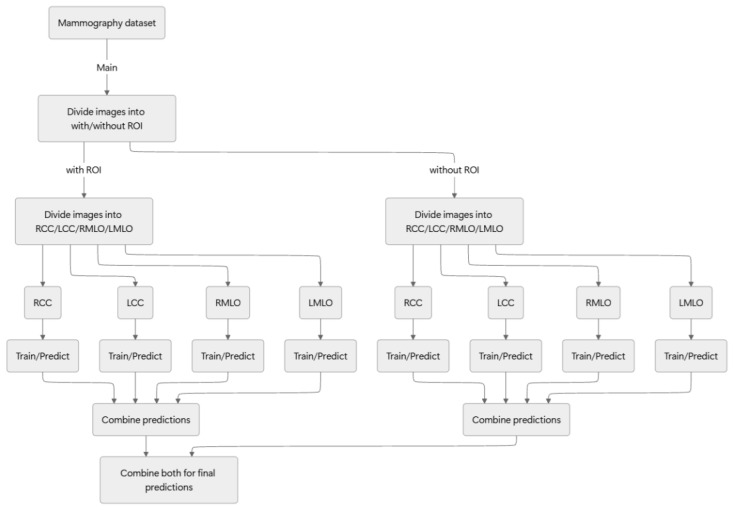
Overview of the proposed method.

**Table 1 bioengineering-12-00885-t001:** Distribution of cases in training and testing subsets of the VinDr dataset.

	Training	Testing	Total
**malignant**	790	198	988
**benign**	4486	1120	5606
**total**	5276	1318	6594

**Table 2 bioengineering-12-00885-t002:** Presence of ROIs in the VinDr dataset.

	with ROIs	Without ROIs	Total
**malignant**	952	36	988
**benign**	816	4790	5606
**total**	1768	4826	6594

**Table 3 bioengineering-12-00885-t003:** Distribution of cases in training and testing subsets of the CDD-CESM dataset.

	Training	Testing	Total
**malignant**	300	31	331
**benign**	300	31	331
**total**	600	62	662

**Table 4 bioengineering-12-00885-t004:** Presence of ROIs in the CDD-CESM dataset.

	with ROIs	Without ROIs	Total
**malignant**	326	5	331
**benign**	322	9	331
**total**	648	14	662

**Table 5 bioengineering-12-00885-t005:** Swin Transformer/Vindr.

	Without Consideration of ROI Mask Image Presence	with Consideration of ROI Mask Image Presence
**Sensitivity**	0.00	0.93
**Specificity**	1.00	0.87
**F-score**	0.00	0.90
**Accuracy**	0.85	0.87

**Table 6 bioengineering-12-00885-t006:** ConvNeXt2/Vindr.

	Without Consideration of ROI Mask Image Presence	with Consideration of ROI Mask Image Presence
**Sensitivity**	0.00	0.90
**Specificity**	1.00	0.86
**F-score**	0.00	0.88
**Accuracy**	0.85	0.87

**Table 7 bioengineering-12-00885-t007:** Swin Transformer/CDD-DESM.

	Without Consideration of ROI Mask Image Presence	with Consideration of ROI Mask Image Presence
**Sensitivity**	0.29	0.65
**Specificity**	0.68	0.65
**F-score**	0.41	0.65
**Accuracy**	0.48	0.65

**Table 8 bioengineering-12-00885-t008:** ConvNeXt2/CDD-DESM.

	Without Consideration of ROI Mask Image Presence	with Consideration of ROI Mask Image Presence
**Sensitivity**	0.65	0.74
**Specificity**	0.65	0.61
**F-score**	0.65	0.67
**Accuracy**	0.65	0.68

## Data Availability

The original contributions presented in this study are included in the article. Further inquiries can be directed to the corresponding author.

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
