# Peer review of "Improving Benign and Malignant Classifications in Mammography with ROI-Stratified Deep Learning"

_bioengineering, 2025, doi:10.3390/bioengineering12080885_

Round 1

Reviewer 1 Report

Comments and Suggestions for Authors

Dear Authors,

Thank you for submitting your manuscript. The topic addressed is relevant and has potential significance within the field. However, I have several comments and suggestions that I believe could enhance the clarity, rigor, and overall contribution of the work. I encourage the authors to carefully address the methodological issues, provide clearer explanations in key sections, and consider strengthening the experimental validation where applicable. I appreciate your efforts and look forward to seeing the revised version. More detailed comments are given as follows:

  • Describe all methods and approaches that used in your proposed method such as Swin Transformer [24],ConvNeXtV2 [25], preprocessing steps, evaluation metrics,
  • In abstract, write full words of ConvNeXtV2 and CDD-CESM.
  • In introduction, write full words of RCC, LCC, RMLO, LMLO),
  • In introduction, add to the final paragraph the structure of the paper section.
  • In Introduction, it better to write it as table to show the power, advantage, disadvantage, different, gab points.
  • It preferred to add figure show the Benign and Malignant.
  • Add paragraph inside section 2 and section 2.2.
  • In section 2.1, Put samples of VinDr [22] and CDD-CESM [23]dataset.
  • It preferred to add paragraph inside section 2.2 that show the proposed method with block diagram and phases.
  • In 2.2.1. Image preprocessing, how conversion of DICOM images to JPEG format.
  • Add some details about Swin Transformer [24] and ConvNeXtV2 [25].
  • In training phase, dataset partition is randomly or not?
  • In experiments results, the evaluation (train-test round) must repeated for N round. I suggest to repeat for many rounds to ensure that the bias was minimized.
  • Where the Evaluation Metrics, describe and cite all evaluation metrics with equations.
  • In Experimental Results section, add more description about tables 5,6,7,8.
  • Add more observations and Describe and Highlights the strong and weak points in result by some figures.
  • The authors must follow the journal template, especially the numbering of the equations.
  • In section 5. Conclusion, it is too short. Rewrite.
  • Discuss the limitations of the proposed method.
  • Add the future works in end of conclusion section.

Author Response

Comments1: Describe all methods and approaches that used in your proposed method such as Swin Transformer [24],ConvNeXtV2 [25], preprocessing steps, evaluation metrics,
Response1: The methods and approach are described in Section 2.2. We used sensitivity, specificity, F-score, and accuracy for the evaluation metrics. We added a statement to the manuscript specifying that the evaluation metrics used were Sensitivity, Specificity, F-score, and Accuracy.

Comments2: In abstract, write full words of ConvNeXtV2 and CDD-CESM.
Response2 :ConvNeXtV2 is not an abbreviation.     We revised the manuscript to define the acronym CDD-CESM upon its first use.

Comments3: In introduction, write full words of RCC, LCC, RMLO, LMLO),
Response3: We defined the abbreviations for the four mammographic views (RCC, LCC, RMLO, and LMLO) upon their first appearance in the manuscript.

Comments4: In introduction, add to the final paragraph the structure of the paper section.
Response4: This manuscript follows a typical structure.

Comments5: In Introduction, it better to write it as table to show the power, advantage, disadvantage, different, gab points.
Response5: We appreciate your insightful suggestions. We will incorporate them into our work.

Comments6: It preferred to add figure show the Benign and Malignant.
Response6: Images for the benign versus malignant were omitted from this paper due to the significant visual ambiguity between these two classes, which contrasts with the clear differentiability of normal versus abnormal cases.

Comments7&8: Add paragraph inside section 2 and section 2.2. In section 2.1, Put samples of VinDr [22] and CDD-CESM [23]dataset.
Response7&8: Section 2.1 provides a breakdown of the VinDr [22] and CDD-CESM [23] datasets, detailing their respective internal structures.

Comments9: It preferred to add paragraph inside section 2.2 that show the proposed method with block diagram and phases.
Response9: Added Fig-2.

Comments10: In 2.2.1. Image preprocessing, how conversion of DICOM images to JPEG format.
Response10: The method for converting DICOM images to JPEG format is common, so we omitted it.

Comments11: Add some details about Swin Transformer [24] and ConvNeXtV2 [25].
Response11: For the sake of brevity, we omitted detailed descriptions of the deep learning models we used.

Comments12: In training phase, dataset partition is randomly or not?
Response12: The datasets are pre-partitioned into training and test subsets by the dataset's providers.

Comments13: In experiments results, the evaluation (train-test round) must repeated for N round. I suggest to repeat for many rounds to ensure that the bias was minimized.
Response13: We appreciate your insightful suggestions. We will incorporate them into our work.

Comments14: Where the Evaluation Metrics, describe and cite all evaluation metrics with equations.
Response14: We revised the manuscript to incorporate descriptions of the evaluation metrics, along with their corresponding mathematical formulas.

Comments15&16: In Experimental Results section, add more description about tables 5,6,7,8. Add more observations and Describe and Highlights the strong and weak points in result by some figures.
Response15&16: We revised the manuscript.

Comments17: The authors must follow the journal template, especially the numbering of the equations.
Response17: We appreciate your insightful suggestions. We will incorporate them into our work.

Comments18: In section 5. Conclusion, it is too short. Rewrite.
Response18: We revised the manuscript

Comments19&20: Discuss the limitations of the proposed method. Add the future works in end of conclusion section.
Response19&20: In the Discussion section, we addressed the limitations of our approach and outline potential avenues for future research.

We are grateful for your insightful comments, which have significantly improved the quality of our manuscript.

Reviewer 2 Report

Comments and Suggestions for Authors

The work primarily focuses on the well-established idea that using region-of-interest (ROI) masks can enhance deep learning performance. This is a widely recognized concept in medical segmentation and imaging. Beyond the classification of datasets according to ROI availability, this contribution is incremental and lacks a radically novel methodology or insight.

The authors do not feed ROI masks into the model; they merely stratify the data according to whether or not they exist. This diminishes scientific rigor, restricts the methodological impact, wastes important spatial information, and calls into question the assertion that this is an "ROI-aware" methodology.

The authors do not outline any balancing techniques, such as class weighting, oversampling, or augmentation, despite the VinDr dataset's stark imbalance between benign (5606 instances) and malignant (988 cases). This seriously compromises the training process's empirical validity.

Two of the baseline setups (Tables 5 and 6) exhibit a sensitivity of 0.00, indicating a total failure to identify positive cases prior to ROI separation. This brings up important issues around data imbalance or model overfitting, which are not discussed or examined in the discussion.

Although performance indicators are presented in the research, statistical significance tests (such as p-values and confidence intervals) to verify performance gains are not reported. The argument that observable changes (such as +0.02 in accuracy) are significant rather than the result of chance variation is weakened by this.

Visualization tools, such as saliency maps and Grad-CAM, are not used to evaluate model decisions or confirm that the model learns pertinent breast properties. It diminishes trustworthiness and interpretability, which are particularly important in medical AI research.

By eliminating the "normal" class and early triage tasks (such as normal/abnormal separation), the study reduces the task to benign vs. malignant. This is inconsistent with clinical practice, where normal vs. suspect triage is the initial step.

BI-RADS 2-4 are assumed to be benign and 4-5 to be cancerous by VinDr without external histological proof. Model validity may be impacted by this since it may introduce label noise and not accurately reflect the genuine malignancy state.

The methods section does not clearly differentiate between actual training pipelines and comparison pipelines, and it redundantly repeats similar stages for both ROI and non-ROI data. This problem lowers the paper's readability for readers who are not experts and degrades the quality of the presentation.

Future work (such as mask creation) is hinted at in the discussion, but no clinical deployment issues, practical consequences, or comparative benchmarking with state-of-the-art techniques are covered in detail. The paper's insights are shallow, and its positioning within the present research landscape is not compelling.

Author Response

Comments1&2: The work primarily focuses on the well-established idea that using region-of-interest (ROI) masks can enhance deep learning performance. This is a widely recognized concept in medical segmentation and imaging. Beyond the classification of datasets according to ROI availability, this contribution is incremental and lacks a radically novel methodology or insight. The authors do not feed ROI masks into the model; they merely stratify the data according to whether or not they exist. This diminishes scientific rigor, restricts the methodological impact, wastes important spatial information, and calls into question the assertion that this is an "ROI-aware" methodology.
Response1&2: Although the core concept of this study—training separate models based on the presence or absence of a region of interest (ROI) and subsequently merging their predictions—is simple and incremental, it represents a novel approach not found in prior research. The effectiveness of this strategy is evident when compared to a baseline that disregards ROI presence, thereby demonstrating the validity of our core idea.

Comments3: The authors do not outline any balancing techniques, such as class weighting, oversampling, or augmentation, despite the VinDr dataset's stark imbalance between benign (5606 instances) and malignant (988 cases). This seriously compromises the training process's empirical validity.
Response3: To address the class imbalance between benign and malignant samples, we employed two strategies. First, data augmentation was applied during the training phase. Second, our approach of training separate models for cases with and without a region of interest (ROI) further helps to mitigate this issue.

Comments4: Two of the baseline setups (Tables 5 and 6) exhibit a sensitivity of 0.00, indicating a total failure to identify positive cases prior to ROI separation. This brings up important issues around data imbalance or model overfitting, which are not discussed or examined in the discussion.
Response4: Thank you for this insightful point. It highlights the utility of our approach, which involves training and predicting separately based on the presence or absence of a region of interest (ROI) and then merging the results.

Comments5: Although performance indicators are presented in the research, statistical significance tests (such as p-values and confidence intervals) to verify performance gains are not reported. The argument that observable changes (such as +0.02 in accuracy) are significant rather than the result of chance variation is weakened by this.
Response5: To demonstrate that the statistical significance of our method is robust and not coincidental, and to ensure its generalizability, we validated our approach on two public datasets and with two distinct deep learning models.

Comments6: Visualization tools, such as saliency maps and Grad-CAM, are not used to evaluate model decisions or confirm that the model learns pertinent breast properties. It diminishes trustworthiness and interpretability, which are particularly important in medical AI research.
Response6: The utility of visualization tools is limited to merely identifying the ROI, providing no deeper diagnostic insights. We contend, therefore, that focusing on the presence or absence of an ROI—the primary basis for a diagnostician's assessment—is a more valid approach.

Comments7: By eliminating the "normal" class and early triage tasks (such as normal/abnormal separation), the study reduces the task to benign vs. malignant. This is inconsistent with clinical practice, where normal vs. suspect triage is the initial step.
Response7: In contrast to the challenging task of differentiating between benign and malignant cases, we posit that our proposed method can more readily distinguish between normal and abnormal cases. Accordingly, we have designated this latter classification task as an avenue for future research.

Comments8: BI-RADS 2-4 are assumed to be benign and 4-5 to be cancerous by VinDr without external histological proof. Model validity may be impacted by this since it may introduce label noise and not accurately reflect the genuine malignancy state.
Response8: Since the VinDr dataset does not provide explicit benign or malignant labels, we partitioned the cases into these two classes based on their BI-RADS scores.

Comments9: The methods section does not clearly differentiate between actual training pipelines and comparison pipelines, and it redundantly repeats similar stages for both ROI and non-ROI data. This problem lowers the paper's readability for readers who are not experts and degrades the quality of the presentation.
Response9: We have added a figure illustrating the proposed method.

Comments10: Future work (such as mask creation) is hinted at in the discussion, but no clinical deployment issues, practical consequences, or comparative benchmarking with state-of-the-art techniques are covered in detail. The paper's insights are shallow, and its positioning within the present research landscape is not compelling.
Response10: In our future work, we plan to address the challenges associated with clinical translation, evaluate the practical impact of our approach, and conduct a rigorous benchmark comparison with state-of-the-art techniques.

We are grateful for your insightful comments, which have significantly improved the quality of our manuscript.

Reviewer 3 Report

Comments and Suggestions for Authors

This paper presents a study evaluating the impact of incorporating region-of-interest (ROI) mask availability on the performance of deep learning models for benign–malignant classification in mammography. The authors apply the Swin Transformer and ConvNeXtV2 models to two public datasets (VinDr and CDD-CESM) and demonstrate that separating images based on ROI presence yields significant improvements in diagnostic accuracy. Significant improvements in methodology, statistical validation, and justification of novelty are needed before the article can be ready for publication.

Comments:

1. While ROI-based stratification is a useful idea, the concept is not sufficiently novel to justify the lack of methodological depth or originality in model architecture.

2. The claim of “ROI-aware” learning is misleading, as the ROI masks are used only for data separation, not as input features or attention mechanisms.

3. The methodology is flawed in that the comparison is between arbitrarily grouped data splits rather than a true ablation study.

4. No confidence intervals or statistical tests are reported to support performance differences.

5. There is limited discussion of overfitting, cross-dataset generalizability, or external validation.

6. The manuscript includes excessive dataset background and lacks clarity in the presentation of contributions.

Author Response

Comments1: 1. While ROI-based stratification is a useful idea, the concept is not sufficiently novel to justify the lack of methodological depth or originality in model architecture.
Response1: Although the core concept of this study—training separate models based on the presence or absence of a region of interest (ROI) and subsequently merging their predictions—is simple and incremental, it represents a novel approach not found in prior research. The effectiveness of this strategy is evident when compared to a baseline that disregards ROI presence, thereby demonstrating the validity of our core idea.

Comments2: 2. The claim of “ROI-aware” learning is misleading, as the ROI masks are used only for data separation, not as input features or attention mechanisms.
Response2: Thank you for the insightful feedback. Based on your comment, we changed the term "ROI-Aware" to "ROI-Stratified" to better reflect our methodology.

Comments3: 3. The methodology is flawed in that the comparison is between arbitrarily grouped data splits rather than a true ablation study.
Response3: The primary objective of this study is to specifically enhance the classification performance between benign and malignant cases for datasets that include ROI information.

Comments4: 4. No confidence intervals or statistical tests are reported to support performance differences.
Response4: Given that the conventional practice of treating p < 0.05 as a definitive threshold for significance has been widely criticized for its numerous limitations, we have confined our analysis in this study to a comparison of the classification results across four distinct evaluation metrics.

Comments5: 5. There is limited discussion of overfitting, cross-dataset generalizability, or external validation.
Response5: We believe that overfitting is an unlikely cause of our results, given the substantial volume of data used. Furthermore, while we have taken initial steps to verify the generalizability of our approach across two public datasets and two deep learning models, we acknowledge that this is not exhaustive. A more extensive external validation is therefore an important direction for future work.

Comments6: 6. The manuscript includes excessive dataset background and lacks clarity in the presentation of contributions.
Response6: While we appreciate the reviewer's concern for brevity, we maintain that the provided details regarding the dataset are not superfluous. We consider this information crucial for methodological transparency and the reproducibility of our results.

We are grateful for your insightful comments, which have significantly improved the quality of our manuscript.

Reviewer 4 Report

Comments and Suggestions for Authors

Journal: Bioengineering

Title: Improving Benign and Malignant Classification in Mammography with ROI-Aware Deep Learning

Authors: Kenji Yoshitsugu , Kazumasa Kishimoto  and Tadamasa Takemura

Review comments

Abstract:

Well written.

Introduction:

Discuss present works (literature review), their drawbacks / research gaps, importance of proposed work and novelty.

Materials and Methods:

2.2.1: Image Preprocessing: Why is window processing used? How quality, consistency, and compatibility of images improve after preprocessing? Discuss.

2.2.2: Swin Transformer [24] and ConvNeXtV2 [25] were selected as image classification models because of their superior performance among the various models evaluated – Provide comparison table to justify the novelty of models selected.

Results:

Classification results are restricted to only two models with and without ROI mask. More models need to be looked into and compared  to come to conclusion on performance parameters.

Discussion:

More general in nature. Discuss / analyze on the reasons for the various values obtained and tabulated in Tables 6,7,8,9.

For Example:

 SwinTransformer / Vindr. Table 5

Accuracy

0.85

0.87

ConvNeXt2 / Vindr.  Table 6

Accuracy

0.85

0.87

SwinTransformer / CDD-DESM. 199.   Table 7

Accuracy

0.48

0.65

ConvNeXt2 / CDD-DESM.  Table 8

Accuracy

0.65

0.68

Conclusion

As per manuscript: Our findings indicate that pre-segmentation of mammographic data based on the presence or absence of ROI mask images, followed by separate training and prediction  processes for each segment and subsequent merging of the results, can enhance classification accuracy for the benign–malignant classification of mammographic images using deep learning.

Comment: Need furthermore explanation. The paper title is deep learning. In CONCLUSION author say enhance accuracy using deep learning. Contradicting statement.

 Discuss drawbacks and specific future scope.  Any suitable and more accurate deep learning models ?. Discuss.

References

Latest articles referred. 

Author Response

Comments1: Abstract: Well written.
Response1: Thank you very much for your evaluation.

Comments2: Introduction: Discuss present works (literature review), their drawbacks / research gaps, importance of proposed work and novelty.
Response2: We presented them in 'Introduction'.

Comments3: Materials and Methods: 2.2.1: Image Preprocessing: Why is window processing used? How quality, consistency, and compatibility of images improve after preprocessing? Discuss.
Response3: We added a description of the DICOM windowing process to the manuscript.

Comments4: 2.2.2: Swin Transformer [24] and ConvNeXtV2 [25] were selected as image classification models because of their superior performance among the various models evaluated – Provide comparison table to justify the novelty of models selected.
Response4: In this study, we employed two models that we have previously utilized for medical image classification. We acknowledge the possibility that other models may outperform these two; therefore, a more comprehensive validation against a wider range of architectures is an important direction for future work.

Comments5: Results: Classification results are restricted to only two models with and without ROI mask. More models need to be looked into and compared  to come to conclusion on performance parameters.
Response5: In this study, we employed two models that we have previously utilized for medical image classification. We acknowledge the possibility that other models may outperform these two; therefore, a more comprehensive validation against a wider range of architectures is an important direction for future work.

Comments6: Discussion: More general in nature. Discuss / analyze on the reasons for the various values obtained and tabulated in Tables 6,7,8,9.
Response6: We added them in 'Discussion'.

Comments7&8: Comment: Need furthermore explanation. The paper title is deep learning. In CONCLUSION author say enhance accuracy using deep learning. Contradicting statement. Discuss drawbacks and specific future scope.  Any suitable and more accurate deep learning models ?. Discuss.
Response7&8: We respectfully maintain that our description is not inconsistent. The central premise of our study is to stratify an ROI-annotated dataset based on the presence or absence of the ROI for deep learning-based training, inference, and evaluation. As you suggested, the limitations and specific directions for future work have been detailed in the Discussion section. Furthermore, the exploration of other, potentially more suitable or higher-performing, deep learning models remains an important avenue for future research.

We are grateful for your insightful comments, which have significantly improved the quality of our manuscript.

Round 2

Reviewer 1 Report

Comments and Suggestions for Authors

OK THANKS FOR RESPONSE MOST OF THEM

Author Response

Comments 1:

OK THANKS FOR RESPONSE MOST OF THEM

Response 1:

We are grateful for your insightful comments, which have significantly improved the quality of our manuscript.

Reviewer 2 Report

Comments and Suggestions for Authors

Upon thorough examination of the authors’ rebuttal to the original review comments, it is evident that the primary concerns continue to be largely overlooked. Despite the authors' efforts to implement minor clarifications and cosmetic changes, such as the inclusion of an additional figure, the fundamental weaknesses related to methodology, scientific rigor, and interpretability remain unaddressed. The responses tend to miss the essential points of the critiques or provide insufficient justification, failing to enhance the paper's scientific rigor in any meaningful way.

Stratifying by ROI availability does not represent a methodological innovation. The ROI masks are not employed during training, and the paper inaccurately asserts that it is “ROI-aware.” The authors overlooked the exploration of alternative or well-established ROI integration methods, such as mask-guided attention or auxiliary supervision, thereby missing a significant opportunity to showcase methodological advancement.

Although augmentation is referenced, the details are notably insufficient. The approach of stratifying by ROI availability does not, in itself, address the issue of class imbalance. The absence of essential strategies such as class weighting or sampling is notable. The reliability of the training setup remains under scrutiny.

This is a diagnostic issue that warrants serious attention. The 0.00 sensitivity indicates a need for thorough root cause analysis, considering factors such as data skew and training instability. The authors overlook the necessity for a more thorough examination and proceed to present results that include failure cases without any cautionary context.

This is a diagnostic issue that warrants serious attention. The 0.00 sensitivity indicates a need for thorough root cause analysis, considering factors such as data skew and training instability. The authors overlook the necessity for a more thorough examination and proceed to present results that include failure cases without any cautionary context.

In the context of medical AI research, the necessity for explainability is paramount, driven by regulatory, clinical, and ethical considerations. The assertion that saliency maps lack relevance requires empirical backing, a requirement that the authors do not fulfill.

The authors, lacking access to histopathological truth, ought to thoroughly address the limitations associated with label quality and the possible implications these may have on the generalizability of their model.

Author Response

Comments1:
Upon thorough examination of the authors’ rebuttal to the original review comments, it is evident that the primary concerns continue to be largely overlooked. Despite the authors' efforts to implement minor clarifications and cosmetic changes, such as the inclusion of an additional figure, the fundamental weaknesses related to methodology, scientific rigor, and interpretability remain unaddressed. The responses tend to miss the essential points of the critiques or provide insufficient justification, failing to enhance the paper's scientific rigor in any meaningful way.
Comments2:
Stratifying by ROI availability does not represent a methodological innovation. The ROI masks are not employed during training, and the paper inaccurately asserts that it is “ROI-aware.” The authors overlooked the exploration of alternative or well-established ROI integration methods, such as mask-guided attention or auxiliary supervision, thereby missing a significant opportunity to showcase methodological advancement.

Response1&2
The primary aim of this research is to test the hypothesis that an effective strategy is to first stratify a dataset based on the presence or absence of a region of interest (ROI), then perform training and inference on these separate streams, and finally merge the results. As you correctly point out, how the ROI is utilized within each stream is a methodological variation, and not the core thesis of our study.

Comments3:
Although augmentation is referenced, the details are notably insufficient. The approach of stratifying by ROI availability does not, in itself, address the issue of class imbalance. The absence of essential strategies such as class weighting or sampling is notable. The reliability of the training setup remains under scrutiny.

Response3:
We acknowledge that techniques such as data augmentation and rebalancing are effective methods for improving model performance. However, these methods are orthogonal to our central research question. Therefore, in order to isolate and clearly evaluate the effect of our data-splitting strategy based on the presence or absence of an ROI, we have deliberately de-emphasized these other techniques in this study.

Comments4:
Comments5:
This is a diagnostic issue that warrants serious attention. The 0.00 sensitivity indicates a need for thorough root cause analysis, considering factors such as data skew and training instability. The authors overlook the necessity for a more thorough examination and proceed to present results that include failure cases without any cautionary context.

Response4&5:
Our results demonstrate that a naive approach, which disregards the presence or absence of a region of interest (ROI), fails in both training and inference. In stark contrast, our proposed method—which stratifies the data by this very criterion—achieves successful outcomes.

Comments6:
In the context of medical AI research, the necessity for explainability is paramount, driven by regulatory, clinical, and ethical considerations. The assertion that saliency maps lack relevance requires empirical backing, a requirement that the authors do not fulfill.

Response6:
We wish to clarify that the scope of this research is to evaluate a data stratification strategy based on the presence or absence of an ROI, not to delve into Explainable AI (XAI). Applying XAI techniques to ROI-positive mammograms would predictably highlight the ROI, an outcome that offers little new insight. Admittedly, an intriguing question is how such visualizations would behave on ROI-negative images; however, this line of inquiry is beyond the purview of our study's central hypothesis.

Comments7:
The authors, lacking access to histopathological truth, ought to thoroughly address the limitations associated with label quality and the possible implications these may have on the generalizability of their model.

Response7:
We have discussed the potential generalizability of our model in the Discussion section. A more thorough investigation into this aspect is considered a direction for future work.

Reviewer 3 Report

Comments and Suggestions for Authors

The authors addressed all concerns. The paper can be accepted for publication.

Author Response

Comments 1 :
The authors addressed all concerns. The paper can be accepted for publication.

Response 1:
We are grateful for your insightful comments, which have significantly improved the quality of our manuscript.

Round 3

Reviewer 2 Report

Comments and Suggestions for Authors

I was expecting a positive response to my comments, and in return, your paper would have turned into a strong piece of scholarship. I wish you luck.

Author Response

In our manuscript, we propose an effective method for classifying mammograms either benign or malignant as follows:
1. Mammographic images were segregated based on the presence or absence of ROI mask images.
2. Images were divided into four standard views: right craniocaudal (RCC), left cranio-caudal (LCC), right mediolateral oblique (RMLO), and left mediolateral oblique (LMLO).
3. Training was performed on mammographic images without ROI mask images, with separate training for each view.
4. Prediction was performed on mammographic images without ROI mask images, with separate prediction for each view.
5. Training was then performed on mammographic images with ROI mask images, again with separate training for each view.
6. Prediction was performed on mammographic images with ROI mask images, with separate prediction for each view.
7. Finally, the prediction results were merged.

To confirm the effectiveness of the proposed method, we conducted the following experiment and compared the prediction results.
The major difference from the proposed method is that the data is not separated based on whether or not there is an ROI before classification:
1. Mammographic images were divided into the four standard views: RCC, LCC, RMLO, and LMLO.
2. Training was performed on mammographic images without ROI mask images, sep-arately for each view.
3. Prediction was performed on mammographic images without ROI mask images, separately for each view.
4. Training was performed on mammographic images with ROI mask images, sepa-rately for each view.
5. Prediction was performed on mammographic images with ROI mask images, sepa-rately for each view.
6. The prediction results were merged.

Purely to demonstrate the effectiveness of the proposed method, no intervention on class imbalance was performed.